# Association of Anemia with Clinical Symptoms Commonly Attributed to Anemia—Analysis of Two Population-Based Cohorts

**DOI:** 10.3390/jcm12030921

**Published:** 2023-01-24

**Authors:** Gesine Weckmann, Simone Kiel, Jean-François Chenot, Aniela Angelow

**Affiliations:** 1Faculty of Applied Health Sciences, European University of Applied Sciences, 18057 Rostock, Germany; 2Department of General Practice, Institute for Community Medicine, University Medicine Greifswald, 17475 Greifswald, Germany

**Keywords:** anemia, symptoms, fatigue, lack of energy, lack of concentration, dyspnea, depression, cardiac failure, COPD, cardiopulmonary disease

## Abstract

Background: Fatigue, dyspnea, and lack of energy and concentration are commonly interpreted as indicative of symptomatic anemia and may thus play a role in diagnostic and therapeutic decisions. Objective: To investigate the association between symptoms commonly attributed to anemia and the actual presence of anemia. Methods: Data from two independent cohorts of the Study of Health in Pomerania (SHIP) were analyzed. Interview data, laboratory data, and physical examination were individually linked with claims data from the Association of Statutory Health Insurance Physicians. A complete case analysis using logistic regression models was performed to evaluate the association of anemia with symptoms commonly attributed to anemia. The models were adjusted for confounders such as depression, medication, insomnia, and other medical conditions. Results: A total of 5979 participants (53% female, median age 55) were included in the analysis. Of those, 30% reported fatigue, 16% reported lack of energy, 16% reported lack of concentration, and 29% reported dyspnea and/or weakness. Anemia was prevalent in about 6% (379). The symptoms were more prevalent in participants with anemia. However, participants with anemia were older and had a poorer health status. There was no association in multivariate logistic regression models between the symptoms fatigue, lack of concentration, dyspnea, and/or weakness and anemia. Anemia was associated (OR: 1.45; 95% CI: 1.13–1.86) with lack of energy in the multivariate analysis. Other factors such as depression, insomnia, and medication were more strongly associated with the symptoms. Conclusion: The clinical symptoms commonly attributed to anemia are unspecific and highly prevalent both in non-anemic and anemic persons. Even in the presence of anemia, other diagnoses should be considered as causes such as depression, heart failure, asthma, and COPD, which are more closely associated with the symptoms. Further diagnostic research is warranted to explore the association of symptoms in different subgroups and settings in order to help clinical decision making.

## 1. Introduction

Anemia is a global public health problem [1] and is associated with morbidity, frailty, and mortality, including maternal and perinatal mortality [1,2,3]. The presence of anemia is an important health indicator, and its detection and treatment can potentially improve overall health [2]. Anemia has been associated with reduced quality of life, especially in people over the age of 60, which is postulated to be due to secondary effects like fatigue and a general reduction of functioning [4,5]. The causality of these associations is still debated and may be partly explained by the association of anemia with multimorbidity and inflammatory conditions [4]. Anemia is often diagnosed in routine laboratory testing as an incidental finding [6,7,8]. Symptoms of anemia described in the literature are fatigue, reduced cognitive function, dyspnea, lack of energy, weakness, and dizziness [1,5,9]. These symptoms can be interpreted as indicative of symptomatic anemia and may thus play a role in diagnostic and therapeutic decisions [10,11,12]. However, little research has been conducted on the association of the symptoms and the presence of anemia, especially when correcting for other possible causal factors.

The aim of this study was to assess the association of anemia with clinical symptoms commonly attributed to anemia in order to provide a broader base of evidence for clinicians when deciding on diagnostic work-up and on prioritizing treatment.

## 2. Materials and Methods

### 2.1. Design and Sample

This is a cross-sectional analysis using data from a prospective longitudinal cohort study in northern Germany (Study of Health in Pomerania) [13]. Data from two independent population-based SHIP cohorts (SHIP and SHIP-TREND) including data on demography, standardized laboratory measurements, and self-reported data from a computer-assisted interview were analyzed. The SHIP cohorts have been described in more detail elsewhere [13,14]. Ambulatory billing data (ICD-10 diagnoses (German modification of the 10th revision of the International Classification of Diseases)) from the Association of Statutory Health Insurance Physicians Mecklenburg-Vorpommern were individually linked with SHIP data. All participants from the second follow-up of the SHIP cohort (SHIP-2, investigation period 2008–2012; N = 2333, response to baseline 54%) and the SHIP-TREND cohort (investigation period 2008–2012, N = 4420) were eligible for inclusion [14]. Participants without billing data or informed consent for the use of billing data, with missing interview data or missing hemoglobin measurements and participants with the diagnosis chronic fatigue syndrome (ICD-Code G93.3) within 5 years prior to the SHIP-2/SHIP-TREND examination excluded from analysis (Figure 1).

### 2.2. Outcomes, Predictors, and Statistical Analysis

The dropout of participants between baseline and the second follow-up was considered by using inverse probability weighting (ipw). To calculate ipw, a logistic regression was performed with study participation as outcome and drop-out predicting variables from baseline investigation (age, sex, diabetes, heart failure, and stroke). The participation probability for follow-up examinations was assigned to each subject, and the reciprocal of this probability was considered in the analysis.

The outcomes fatigue, lack of energy, lack of concentration, dyspnea, and/or weakness were self-reported data from a self-assessment questionnaire. The answer categories none, mild, moderate, and severe were given for the outcomes fatigue, lack of energy, and lack of concentration. We dichotomized these variables into ‘not present’ (none and mild) and ‘present’ (moderate and severe). The answer categories for dyspnea and/or weakness were ‘only dyspnea’, ‘only weakness’, ‘both’, or ‘neither dyspnea nor weakness’. Anemia and anemia severity were defined according to the hemoglobin levels introduced by the World Health Organization (WHO). Thus, a hemoglobin level <120 g/L in women, <130 g/L in men, and <110 g/L in pregnant women was defined as anemia [15]. Iron deficiency was defined as serum ferritin <8 µg/L for women and <26 µg/L for men. Pregnancy was defined as beta-HCG ≥ 5 IU/L.

The variables extracted from billing data are defined in Table 1. Hypertension was defined as >140/90 mmHg of the average of the second and third blood pressure measurement at SHIP examination. Participants who reported moderate to severe depression, low mood, or anxiety were considered as having depression or low mood and/or anxiety. Insomnia was defined as having difficulties falling asleep more than 3 times a week (SHIP-Trend) or often (SHIP-2) or difficulties sleeping through the night (interview data). Information on medical history was obtained using the computer-assisted interview. Medications with fatigue as a side effect were identified according to the fatigue guideline by the German Society for General Practice and Family Medicine (DEGAM) [16] and included cytotoxic agents, benzodiazepines, antidepressants, antipsychotics, antihistamines, antihypertensive medications, opioids, and anti-Parkinson medications (Table 2). The Charlson Comorbidity Index (CCI) was calculated on the basis of billing diagnoses, either acute or permanent, within 1 year prior to enrollment in SHIP-2/SHIP-Trend [17].

Continuous hemoglobin levels (g/L) according to the presence of symptoms were depicted using box plots. Univariate, bivariate, and multivariate logistic regression models were applied to assess the association between anemia and the outcomes fatigue, lack of energy, lack of concentration, dyspnea, and/or weakness. We used stepwise selection to select the most suitable variables in the models. Multicollinearities among predictor variables in the same model were tested, with all correlations being <0.6. The variables ‘number of medications’ and ‘medication with fatigue as a side-effect’ could not be included in the same model because they were correlated. A sensitivity analysis was performed to determine the association of iron deficiency with anemia symptoms. We included the analysis in Appendix A. To assess if the composite of anemia symptoms was predictive of anemia, positive predictive values (PPV) of the presence of one or more anemia symptoms for the presence of anemia were calculated. The analyses were performed using SAS Institute Inc., Cary, NC, USA, Software 9.4.

## 3. Results

### 3.1. Characteristics of Study Population

A total of 5979 participants (53% female, median age 55 years) were included in the analysis. Fatigue was reported by 30%, lack of energy and lack of concentration were reported by 16% respectively, and dyspnea and/or weakness by 29% of participants (Table 3). The number of participants suffering from depression and anxiety was higher in self-reported (21%) as opposed to claims data (12%). Insomnia was reported by 24%.

### 3.2. Anemia

Anemia was observed in 379 (6.4%) of participants according to SHIP data. Of those, 48% were female, and the median age was 64 years. Most participants had mild anemia (78% female and 92% male) (Table 4). In participants with anemia, fatigue was reported by 34% (128/379), lack of energy by 21% (81/379), lack of concentration by 19% (70/379), and dyspnea and/or weakness by 38% (145/379). This subgroup was older and in a poorer health condition. A total of 25% had diabetes, 12% had chronic kidney disease, 7% had heart failure, and 7% had chronic obstructive pulmonary disease (Table 3). The CCI was higher among participants with anemia, as well as the number of prescribed medications and the number of medications with fatigue as a side effect. According to claims data, anemia was documented in 2.4% (43/5979) of the study population.

### 3.3. Association of Anemia and Symptoms

#### 3.3.1. Fatigue

Of those who reported fatigue (30%), 7% (128/1775) had anemia. The univariate logistics regression model of anemia and fatigue revealed a weak association, which did not reach significance (OR = 1.19 (95% CI: 0.99–1.44)). The median hemoglobin levels of participants with fatigue were slightly lower than the hemoglobin levels of participants without fatigue. Overall, there was no difference in hemoglobin level distribution between participants with and without fatigue (Figure 2). In the bivariate analysis, fatigue was significantly associated with anemia after adjusting for age, sex, depression, and/or anxiety (Appendix A). In the multivariate analysis, there was no association between anemia and fatigue (Table 5). Furthermore, fatigue was not associated with iron deficiency (Appendix A).

#### 3.3.2. Lack of Energy

Of those who reported lack of energy (16%), 8% (81/968) had anemia. In the univariate model, anemia and lack of energy were associated with an OR = 1.38 (95% CI: 1.11–1.70). The distribution of hemoglobin levels between participants with lack of energy compared to participants without this symptom was similar (Figure 2). In the bivariate analysis, lack of energy was significantly associated with anemia after adjusting for age, sex, hypothyroidism, depression and/or anxiety, insomnia, CCI, cancer, heart failure, diabetes, and pregnancy (Appendix A). After adjusting in the multivariate model, the odds ratios for the association of anemia and lack of energy were 1.41 (95% CI: 1.09; 1.81) and 1.45 (95% CI: 1.13; 1.86), respectively (Table 5). No association was observed between lack of energy and iron deficiency (Appendix A).

#### 3.3.3. Lack of Concentration

Of those who reported lack of concentration (16%), 8% (70/927) had anemia. In the univariate analysis, anemia and lack of concentration were associated with an OR = 1.29 (95% CI: 1.03–1.61). Participants with the symptom lack of concentration had slightly lower median hemoglobin levels compared to participants without lack of concentration. Overall, the distribution of hemoglobin levels did not differ between the groups of participants with and without lack of concentration (Figure 2). In the bivariate analysis, anemia was associated with lack of concentration after adjusting for sex, hypothyroidism, depression and/or anxiety, cancer, and pregnancy (Appendix A). In multivariate analysis, no significant association between anemia and lack of concentration was found (Table 5), but iron deficiency was significantly associated with lack of concentration with an OR of 1.65 (95%CI 1.24–2.19) (Appendix A).

#### 3.3.4. Dyspnea

Of those reporting dyspnea and/or weakness (30%), 8% (145/1727) had anemia. In the univariate analysis, anemia was associated with dyspnea and/or weakness with an OR = 1.55 (95% CI: 1.29–1.85). Participants with these symptoms had slightly lower median hemoglobin levels than participants without the symptoms. Overall, no difference in the distribution was observed (Figure 2). In the bivariate analysis, the association remained significant (Appendix A). In the multivariate analysis, no association between anemia and dyspnea and/or weakness was observed (Table 5), and no association was observed between iron deficiency and dyspnea and/or weakness (Appendix A).

### 3.4. Positive Predictive Value of Symptoms for the Presence of Anemia

In participants with anemia, 6.6% had all four symptoms compared to 4.7% of participants without anemia (Table 6). The positive predictive value was 8% for all four symptoms and 6% for the presence of one symptom.

### 3.5. Association of Symptoms with Other Variables

In the bivariate analysis, the predictors female, medication with fatigue as a side-effect, depression/anxiety, and insomnia were strongly associated with the outcomes fatigue, lack of energy, and lack of concentration (Figure 3, Figure 4 and Figure 5). The predictors heart failure, COPD, asthma, chronic kidney disease, medication with fatigue as a side-effect, and depression were strongly associated with the outcome dyspnea and/or weakness (Figure 6).

In the multivariate analysis, the predictor depression/anxiety had the strongest association with all anemia symptoms, especially with lack of energy. Number of medications, medication with fatigue as a side effect, insomnia, and an increased CCI were associated with all anemia symptoms. Heart failure, asthma, and COPD were strongly associated with dyspnea and/or weakness (Table 5).

## 4. Discussion

### 4.1. Summary of the Main Results

The prevalence of anemia in the study population was 6%. The majority of anemic subjects were classified as having mild anemia (78–92%).

Participants with anemia were older and had a higher prevalence of chronic conditions such as diabetes and chronic kidney disease, as well as a higher CCI than participants without anemia. The number of medications was higher in participants with anemia, and they used more medications with fatigue as a possible side effect. Anemia was associated with lack of energy in the multivariate logistic regression. No association was found between fatigue, lack of concentration, dyspnea, and/or weakness with anemia after adjusting for confounders, but lack of concentration was weakly associated with iron deficiency. The strongest association with anemia symptoms was found for depression and/or anxiety with an odds ratio ranging from 2.68 to 8.22. Furthermore, the symptoms were associated with polypharmacy, medications with fatigue as a possible side effect, insomnia, and CCI. Heart failure, COPD, and asthma were associated with dyspnea and/or weakness. The positive predictive value of four anemia symptoms for the presence of anemia was 8%.

### 4.2. Comparison with Scientific Literature

Our findings on predictivity of anemia symptoms are congruent with research by Wood (1966), who found no association between symptoms that are attributed to anemia and the presence of anemia in a survey among community dwelling adults in Great Britain [18]. In a meta-analysis of the etiology of tiredness as a symptom in general practice, the prevalence of anemia ranged from 0 to 6.9% [19]. This is in accordance with our findings, where 7% of study participants complained about fatigue and had anemia. Furthermore, they found the most common diagnosis of patients presenting with tiredness in primary care was depression, with 18.5% (95% CI: 16.2–21.0%) being diagnosed as such [19]. This is in accordance with our findings, where the strongest association of anemia symptoms was found for depression and/or anxiety. In a Dutch study among patients presenting with tiredness in general practice, Knottnerus (1986) reported no significant difference among patients and controls regarding anemia prevalence [20]. In contrast, Macher et al. (2020) reported a significant reduction in symptoms of fatigue after iron supplementation in a study among iron-deficient blood donors, but this study was neither blinded nor did it contain a control group of subjects without iron deficiency, or a placebo group [21]. Some research suggests that iron deficiency rather than hemoglobin levels may be the causal factor in some of the anemia symptoms, and that these symptoms may be alleviated by iron supplementation [22,23,24]. We found iron deficiency as indicated by low ferritin levels to be correlated with lack of concentration, but not with other anemia symptoms, in the multivariate analysis.

It is possible that the positive predictive value of anemia symptoms is higher in populations with a high a priori risk of anemia, e.g., cancer patients, who typically have high rates of anemia, ranging from 30 to 90% [25]. This is confirmed by Johnstone et al. (2020), who found a PPV of 80% for tiredness and dyspnea in a study among cancer patients, albeit with a low sensitivity of 8% [25]. Robalo Nunes et al. (2020) reported a 20% higher probability of anemia in persons with symptoms, but they recruited subjects in healthcare settings and had an extraordinarily high rate of anemia of up to 52%, which is not representative of the general population as reported by other sources, and the purported symptoms included bleeding and visible blood loss, which are causal risk factors, rather than symptoms [26].

Anemia was more prevalent in participants over the age of 50 and in participants with chronic diseases such as CKD and diabetes, leading to an increased number of medications and therefore an increased number of medications with the possible side effect of fatigue. General screening for anemia is not recommended but is usually part of CKD monitoring or prenatal care and is frequently part of standard laboratory testing [6,7]. Clinicians should consider elderly, fragile individuals and individuals with chronic diseases as a higher risk subgroup when testing for anemia.

### 4.3. Limitations

We conducted a complete case analysis. Due to the large sample size and the low number of excluded subjects, we do not expect a relevant influence on our results. When interpreting the results, it should be considered that most study participants had mild anemia. The low number of subjects with severe anemia did not allow further analysis of this subgroup. Additionally, SHIP cohort subjects are almost exclusively Caucasian, and thus certain genetic forms of anemia, such as hemoglobinopathy, are underrepresented. Due to the study setting, the conclusions cannot be extrapolated to in-patient clinical settings where the a priori risk of anemia is considerably higher. It is likely that the positive predictive value of anemia symptoms can be different in an in-patient population. Some research suggests that iron deficiency rather than hemoglobin levels may be the causal factor in some of the anemia symptoms and that these symptoms may be alleviated by iron supplementation [21,22,24]. In this study, we were limited to the analysis of laboratory values that were available in the SHIP-study, which included ferritin as an indicator of iron storage, but did not include hepcidin or transferrin saturation as additional indicators of iron metabolism.

### 4.4. Interpretation

Patients with chronic anemia often show surprisingly little symptoms, and anemia is often discovered coincidentally due to routine diagnostic testing. Although anemia has been associated with lower quality of life and several unfavorable health outcomes, this does not necessarily prove a causal relation, because of confounding factors [4].

Physicians in primary and ambulatory care often rely predominantly on patient history and symptoms when deciding on differential diagnosis and diagnostic testing. This analysis has shown that the diagnostic value of anemia symptoms is very low in the general population, which is probably inherent to the unspecific nature of the symptoms and the low prevalence of severe anemia in this population. Our data show that diagnostic work-up in patients with symptoms such as fatigue or dyspnea should not be limited primarily to anemia but rather focus on conditions such as depression, heart failure, asthma, and COPD. Because of the relatively high incidence of anemia in persons over the age of 60, age can be a useful indicator when deciding on diagnostic testing. Additionally, there is a rationale for screening and treatment of anemia in persons suffering from cardiac or pulmonary symptoms, in order to improve functional capacity and potentially quality of life.

## 5. Conclusions

The clinical symptoms commonly attributed to anemia are unspecific and highly prevalent in non-anemic and anemic persons alike. Diagnostic work-up in patients with symptoms such as fatigue, lack of energy and concentration, or dyspnea should not primarily focus on anemia. Other possible causes such as depression, heart failure, asthma, and COPD are more likely and should be given more consideration, especially in out-patient settings.

## Figures and Tables

**Figure 1 jcm-12-00921-f001:**
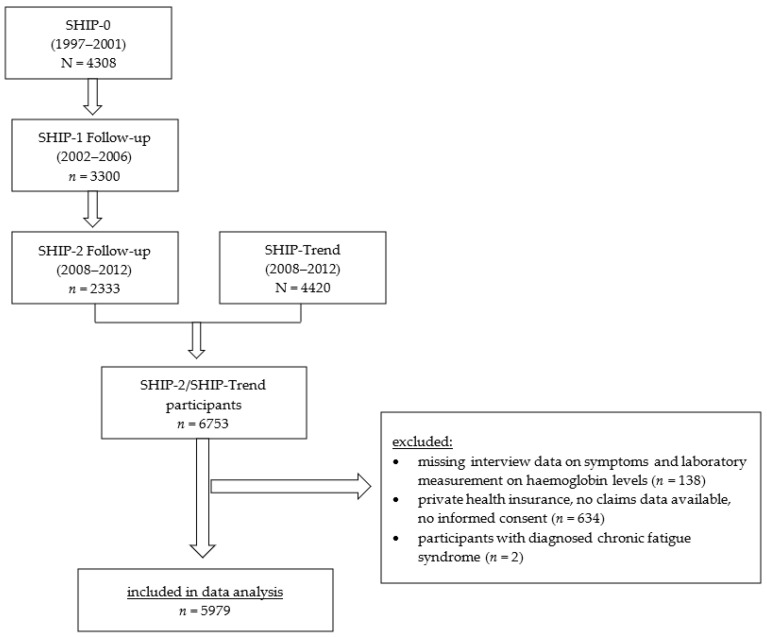
Flow chart of study population. SHIP: Study of Health in Pomerania.

**Figure 2 jcm-12-00921-f002:**
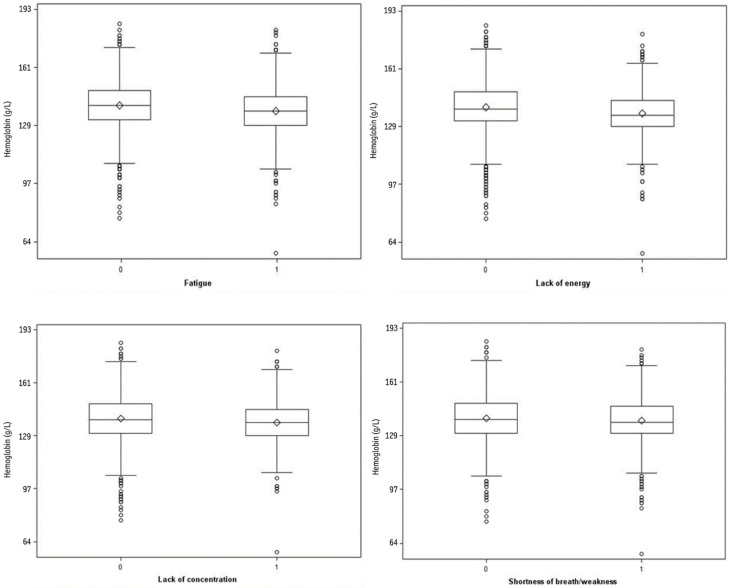
Box plot depiction of continuous hemoglobin levels (g/L) in study subjects according to the presence of the symptoms fatigue, lack of energy, lack of concentration, and dyspnea and weakness (0 = symptom is not present; 1 = symptom is present).

**Figure 3 jcm-12-00921-f003:**
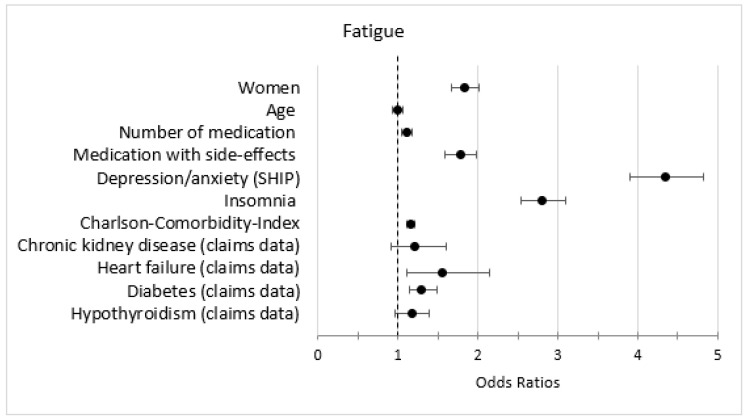
Bivariate logistic regression models (anemia + second variable) with the outcome fatigue; only the estimates (odds ratios and 95% confidence intervals) of the second variable is shown, (weighted).

**Figure 4 jcm-12-00921-f004:**
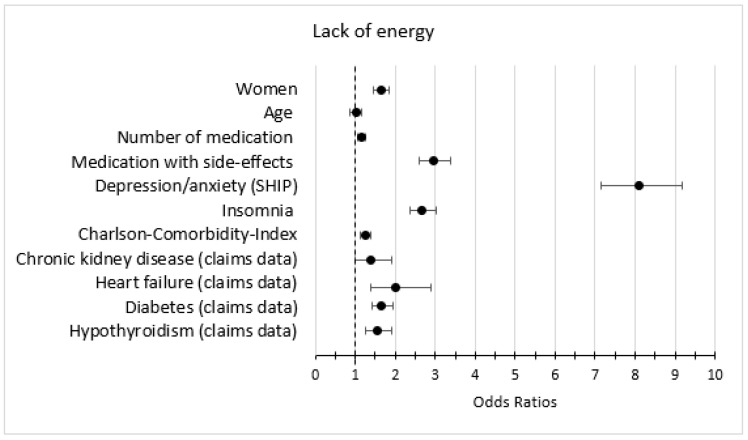
Bivariate logistic regression models (anemia + second variable) with the outcome lack of energy; only the estimates (odds ratios and 95% confidence intervals) of the second variable is shown (weighted).

**Figure 5 jcm-12-00921-f005:**
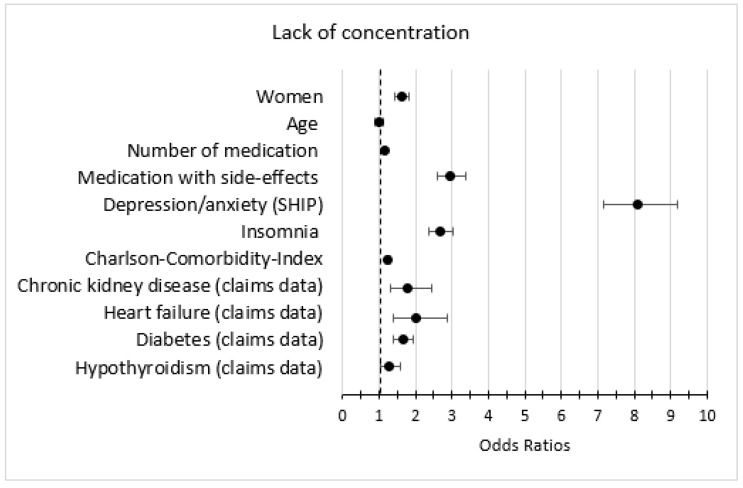
Bivariate logistic regression models (anemia + second variable) with the outcome lack of concentration; only the estimates (odds ratios and 95% confidence intervals) of the second variable is shown (weighted).

**Figure 6 jcm-12-00921-f006:**
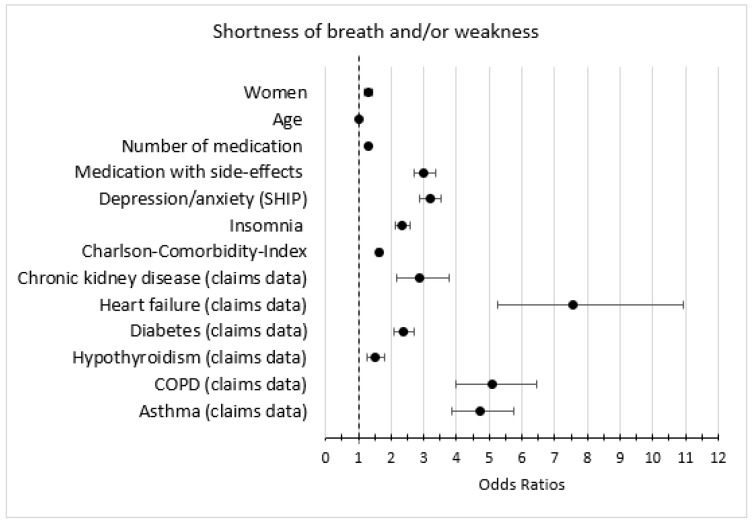
Bivariate logistic regression models (anemia + second variable) with the outcome dyspnea and/or weakness; only estimates (odds ratios and 95% confidence intervals) of the second variable is shown (weighted).

**Table 1 jcm-12-00921-t001:** Definitions of diseases according to claims data.

Disease	Billing Diagnosis ICD-10 GM	Time Frame and Kind of Diagnosis
		at least one relevant and confirmed ICD diagnosis coded as …. prior to the SHIP-2/SHIP-Trend study entrance of the participants
Anemia	D50, D64, D55, D59, D53, D51, D61	acute diagnosis within 1 year
Heart Failure	I50	acute diagnosis within 1 year permanent diagnosis within 5 years
Depression, Anxiety	F32, F33, F41, F31, F06, F20, F61, F40, F43, F34, F60	acute diagnosis within 1 year
Diabetes	E10, E11, E13, E14, E12	acute diagnosis within 1 year permanent diagnosis within 5 years
Chronic Kidney Disease	N18	acute diagnosis within 1 year permanent diagnosis within 5 years
Hypothyroidism	E01, E02, E03	acute diagnosis within 1 year permanent diagnosis within 5 years
COPD	J44	acute diagnosis within 1 year permanent diagnosis within 5 years
Asthma	J45	acute diagnosis within 1 year permanent diagnosis within 5 years
Cancer	C0–C96	acute or permanent diagnosis within 1 year
Charlson Comorbidity Index	relevant diagnosis	acute or permanent diagnosis within 1 year

ICD: International Classification of Diseases, COPD: chronic obstructive pulmonary disease, SHIP: Study of Health in Pomerania.

**Table 2 jcm-12-00921-t002:** Medication with fatigue as a possible side effect and their Anatomical Therapeutic Chemical Classification (ATC) according to the guideline ‘fatigue’ by the German Society for General Practice and Family Medicine (DEGAM) [16].

Medication	ATC-CODES
Antineoplastic Agents	L01
Benzodiazepine Derivatives	N05CD
Antidepressants	N06A
Antipsychotics	N05A
Antihistamine	R06A
Antihypertensives and Diuretics	C02, C03
Opioids	N02A
Anti-Parkinson Drugs	N04

**Table 3 jcm-12-00921-t003:** Characteristics of all study participants and participants according to anemia status.

	All Participants (N = 5979; 100%)	Participants with Anemia (*n* = 379; 6.3%)	Participants without Anemia (*n* = 5600; 93.7%)
Women	3149 (52.7)	183 (3.06)	2966 (49.6)
Age			
20–29	327 (5.5)	18 (0.3)	309 (5.2)
30–49	84 (1.4)	1953 (32.7)	2037 (34.1)
50–69	2519 (42.1)	138 (2.3)	2381 (39.8)
>=70	1096 (18.3)	139 (2.3)	957 (16.0)
Fatigue	1775 (29.7)	128 (2.2)	1647 (27.5)
Lack of energy	968 (16.2)	81 (1.4)	887 (14.8)
Lack of concentration	927 (15.5)	70 (1.2)	857 (14.3)
Shortness of breath/weakness	1727 (28.9)	145 (2.4)	1582 (26.5)
Iron deficiency (serum ferritin)	292 (5.0) (180 missing)	95 (1.6) (13 missing)	197 (3.4) (167 missing)
Depression /anxiety (SHIP-data)	1273 (21.3) (22 missing)	74 (1.2) (2 missing)	1199 (20.1) (20 missing)
Depression / anxiety (claims data)	715 (12.0)	45 (0.8)	670 (11.2)
Insomnia	1453 (24.3)	109 (1.8)	1344 (22.5)
Hypertension	569 (9.6) (30 missing)	16 (0.3) (3 missing)	553 (9.3) (27 missing)
Heart failure (claims data)	121 (2.0)	28 (0.4	93 (1.6)
Chronic kidney disease (claims data)	165 (2.8)	44 (0.8)	121 (2.0)
Diabetes (claims data)	794 (13.3)	94 (1.6)	700 (11.7)
Hypothyroidism (claims data)	375 (6.3)	27 (0.5)	348 (5.8)
Cancer (claims data)	92 (1.5)	14 (0.2)	78 (1.3)
Asthma (claims data)	301 (5.0)	20 (0.3)	281 (4.7)
COPD (claims data)	209 (3.5)	28 (0.5)	181 (3.0)
Charlson Comorbidity Index (claims data)			
0	4417 (73.9)	216 (3.6)	4201 (70.3)
1	951 (15.9)	63 (1.0)	888 (14.9)
2	324 (5.4)	42 (0.7)	282 (4.7)
3–5	253 (4.2)	49 (0.8)	204 (3.4)
6–9	34 (0.6)	9 (0.2)	25 (0.4)
Number of medications			
0	1656 (27.7)	69 (1.2)	1587 (26.5)
1	1125 (18.8)	48 (0.8)	1077 (18.0)
2	877 (14.7)	40 (0.7)	837 (14.0)
3	607 (10.2)	37 (0.7)	570 (9.5)
4	465 (7.8)	29 (0.5)	436 (7.3)
5	385 (6.4)	37 (0.6)	348 (5.8)
>6	864 (14.4)	119 (1.9)	745 (12.5)
Medication with possible side-effect fatigue	1061 (17.8)	112 (1.9)	949 (15.9)
Pregnant	199 (3.3)	4 (0.1)	195 (3.2)

COPD: chronic obstructive pulmonary disease, SHIP: Study of Health in Pomerania.

**Table 4 jcm-12-00921-t004:** Categorization of anemia severity in anemic study subjects according to hemoglobin normal values from the World Health Organization [15].

Anemia Severity	Women (*n* = 179)	Pregnant Women (*n* = 4)	Men (*n* = 196)
Mild	110–119 g/L	100–109 g/L	110–129 g/L
	140 (78.4%)	4 (100%)	179 (92.2%)
Moderate	80–109 g/L	70–99 g/L	80–109 g/L
	39 (21.6%)	-	15 (6.4%)
Severe	<80 g/L	<70 g/L	<80 g/L
	-	-	2 (1.4%)

Anemic subjects from the population-based study cohorts SHIP-START-2 and SHIP-TREND were categorized according to WHO normal values for hemoglobin levels in the respective patient groups ‘women’, ‘pregnant women’, and ‘men’. WHO: World Health Organization, SHIP: Study of Health in Pomerania.

**Table 5 jcm-12-00921-t005:** Multivariate logistic regression models with symptoms of anemia as outcome variables (weighted).

	Outcomes							
	Fatigue		Lack of Energy		Lack of Concentration		Dyspnea and/or Weakness	
**Predictors**	**OR (95% CI)** ***n* = 5957**		**OR (95% CI)** ***n* = 5957**		**OR (95% CI)** ***n* = 5957**		**OR (95% CI)** ***n* = 5957**	
	Model 1	Model 2	Model 1	Model 2	Model 1	Model 2	Model 1	Model 2
Anemia	1.15 (0.94; 1.40)	1.19 (0.98; 1.46)	**1.41** **(1.09; 1.81)**	**1.45** **(1.13; 1.86)**	1.12 (0.87; 1.44)	1.13 (0.88; 1.46)	1.04 (0.85; 1.29)	1.11 (0.90; 1.37)
Women	**1.40** **(1.26; 1.55)**	**1.42** **(1.28; 1.57)**	**1.33** **(1.16; 1.52)**	**1.33** **(1.16; 1.53)**	1.12 (0.98; 1.28)	1.11 (0.97; 1.28)	1.08 (0.97; 1.21)	1.09 (0.98; 1.22)
Age (years)	0.98 (0.97; 0.99)	0.98 (0.98; 0.99)	0.98 (0.97; 0.99)	0.99 (0.98; 1.00)	1.01 (1.00; 1.01)	1.01 (1.00; 1.02)	**1.01** **(1.01; 1.02)**	**1.03** **(1.02; 1.04)**
Number of medications	**1.11** **(1.08; 1.13)**	-	**1.13** **(1.09; 1.16)**	-	**1.08** **(1.05; 1.10)**	-	**1.19** **(1.17; 1.22)**	-
Medication with possible side effect fatigue	-	**1.25** **(1.10; 1.42)**	-	**1.58** **(1.36; 1.85)**	-	**1.70** **(1.47; 1.98)**	-	**1.69** **(1.49; 1.92)**
Depression and/or anxiety (SHIP)	**3.44** **(3.08; 3.84)**	**3.49** **(3.13; 3.91)**	**8.22** **(7.22; 9.36)**	**8.13** **(7.14; 9.26)**	**6.79** **(5.95; 7.74)**	**6.58** **(5.77; 7.51)**	**2.66** **(2.36; 2.99)**	**2.68** **(2.38; 3.01)**
Insomnia (SHIP)	**2.24** **(2.01; 2.49)**	**2.31** **(2.08; 2.57)**	**2.06** **(1.80; 2.35)**	**2.12** **(1.86; 2.43)**	**1.76** **(1.54; 2.02)**	**1.78** **(1.56; 2.04)**	**1.65** **(1.47; 1.84)**	**1.72** **(1.54; 1.93)**
CCI (claims data)	**1.10** **(1.04; 1.16)**	**1.18** **(1.12; 1.25)**	1.07 (0.99; 1.14)	**1.15** **(1.08; 1.22)**	1.08 (1.01; 1.15)	**1.11** **(1.04; 1.18)**	**1.09** **(1.03; 1.16)**	**1.20** **(1.14; 1.27)**
Heart failure (claims data)	-	-	-	-	-	-	**2.66** **(1.7; 4.00)**	**3.03** **(2.03; 4.53)**
COPD (claims data)	-	-	-	-	-	-	**1.99** **(1.49; 2.65)**	**2.14** **(1.61; 2.83)**
Asthma (claims data)	-	-	-	-	-	-	**3.36** **(2.67; 4.25)**	**3.63** **(2.88; 4.56)**

OR: odds ratio, 95% CI: 95% confidence interval, CCI: Charlson Comorbidity Index, COPD: chronic obstructive pulmonary disease, SHIP: Study of Health in Pomerania. Bold numbers = statistically significant.

**Table 6 jcm-12-00921-t006:** Number of symptoms in subgroups and positive predictive value of anemia symptoms, weighted percentages.

Number of Symptoms	All Participants (*n* = 5979) (%)	Participants without Anemia (*n* = 5600) (%)	Participants with Anemia (*n* = 379) (%)	Positive Predictive Value % (95% CI)
Four Symptoms	295 (4.9)	271 (4.7)	24 (6.6)	8.4 (5.8–10.9)
Three Symptoms	430 (7.2)	387 (6.9)	43 (10.5)	8.9 (6.8–11.2)
Two Symptoms	708 (11.5)	658 (11.4)	50 (13.1)	6.9 (5.4–8.5)
One Symptom	1511 (24.7)	1412 (24.7)	99 (24.6)	6.1 (5.1–7.1)

CI: confidence interval.

## Data Availability

Data from the SHIP study are available on reasonable request, in accordance with German law and the data protection concept of the Institute for Community Medicine https://www.fvcm.med.uni-greifswald.de/dd_service/data_use_intro.php.

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
