# Peer review of "Association of Anemia with Clinical Symptoms Commonly Attributed to Anemia—Analysis of Two Population-Based Cohorts"

_jcm, 2023, doi:10.3390/jcm12030921_

Round 1
Reviewer 1 Report
I thank the Editor for the opportunity to review the manuscript tilted 'Association of anemia with clinical symptoms commonly attributed to anemia – analysis of two population-based cohorts'.
The Authors aimed to analyze the association between symptoms commonly attributed to anemia with its presence, based on 2 historic population-based cohorts.
I praise the authors for the effort taken to perform such a robust analysis. However, I have some remarks.
In the Background section the Authors stated that unspecific symptoms like 'fatigue, shortness of breath, lack of energy and concentration' play a pivotal role for anemia diagnosis. The established method for anemia diagnosis is Hb concentration determination, not these unspecific self-declared symptoms that may accompany numerous diseases other than anemia. This statement requires modification.
The authors should be consistent with the use of terms. I suggest that the term dyspnea (a keyword) is used throughout the manuscript.
The Authors should clarify why they chose to analyze two cohorts from the same time period? In what way were these cohorts different? This point requires more clarification.
The authors analyzed the research question based on historic population data from 2008-2012. This should be listed as a limitation, as the results may not be directly extrapolated to the current out-patient population.
Why Figure 2 is in the Materials and Methods section? Surely it should appear in the Results. Description of Fig. 2 in the main text and the figure legend should be more precise. Please change Hb concentration units to g/L throughout the manuscript as it is more common in the subject literature.
I do not understand why the Authors decided to compare all participants (including participants with anemia) to participants with anemia in Table 3. In my view participants with and without anemia should be compared and p values for intergroup differences shown. Alternatively there should be 3 columns in the Table: all participants; participants without anemia; participants with anemia. Percents of participants in both groups are missing at the top of the Table 3.
Legend for Table 4 requires correction.
In line 144 please give Me (IQR) and p value for age in both groups.
Table 5- please explain what 2 models of multivariate logistic regression was used. This should also be clarified in the Materials and Methods. Why are there empty cells?
In the 'Comparison with scientific literature' there is no mention of papers showing different prevalence of symptoms of anemia in anemia of various severity. It is important to stress this as participants in the study were mostly mildly anemic.
Citing the paper from 1966 is concerning me. The cited references should be recent publications - optimally from within the last 5 years.
In conclusions the Authors should stress that anemia in these cohorts was mostly mild, occasionally moderate.
The English language style should be improved in some parts of the manuscript. I suggest that the manuscript is read by a native speaker.
Author Response
- In the Background section the Authors stated that unspecific symptoms like 'fatigue, shortness of breath, lack of energy and concentration' play a pivotal role for anemia diagnosis. The established method for anemia diagnosis is Hb concentration determination, not these unspecific self-declared symptoms that may accompany numerous diseases other than anemia. This statement requires modification.
We agree, that cases of anemia are usually detected in laboratory results. This is true for clinical and ambulatory settings. On the other hand, in primary care settings, patients often present with vague or unspecific complaints. In these settings, the fact that e.g. fatigue is more closely correlated with depression than with anemia, can be helpful in determining on the direction of diagnostic effort and additionally we found the mention of symptomatic anemia in guidelines, when deciding on management.
We have changed the manuscript according to your recommendation from:
“Symptoms of anemia described in the literature are fatigue, reduced cognitive function, dyspnea, lack of energy, weakness and dizziness [1, 5, 6]. These symptoms can be interpreted indicative of symptomatic anemia and may thus play a substantial role in diagnostic and therapeutic decisions [7, 8, 9]. However, little research has been conducted on the association of the symptoms and the presence of anemia, especially when correcting for other possible causal factors.”
To:
“Anemia is often diagnosed in routine laboratory testing as an incidental finding [6,7,8]. Symptoms of anemia described in the literature are fatigue, reduced cognitive function, dyspnea, lack of energy, weakness and dizziness [1, 5, 6]. These symptoms can be interpreted as indicative of symptomatic anemia and may thus play a substantial role in diagnostic and therapeutic decisions [7, 8, 9]. However, little research has been conducted on the association of the symptoms and the presence of anemia, especially when correcting for other possible causal factors.”
- The authors should be consistent with the use of terms. I suggest that the term dyspnea (a keyword) is used throughout the manuscript.
The manuscript was changed, and “shortness of breath” was replaced by “dyspnea”.
- The Authors should clarify why they chose to analyze two cohorts from the same time period? In what way were these cohorts different? This point requires more clarification.
We decided to analyze two independent cohorts to have a larger subject base for the analysis. The SHIP-START and SHIP-TREND cohorts are comparable and have been described in detail elsewhere*. We have added a reference in the methods section of this manuscript, to highlight this.
*Völzke H, Schössow J, Schmidt CO, et al. Cohort Profile Update: The Study of Health in Pomerania (SHIP). International Journal of Epidemiology 2022: dyac034. https://doi.org/10.1093/ije/dyac034
- The authors analyzed the research question based on historic population data from 2008-2012. This should be listed as a limitation, as the results may not be directly extrapolated to the current out-patient population.
The population-based study data were gathered in the period 2008-2012. We do not expect the study period to influence the correlation between symptoms and the presence of anemia, as this correlation will be physiological and not time-dependent. The following text was added to the “Limitations” section:
“This analysis was conducted in a population-based cohort during the years 2008-2012. Although we do not expect this to have influenced the results, because we expect the correlation between symptoms and anemia to be physiological and not time-dependent, this should be considered when interpreting the results.”
- Why Figure 2 is in the Materials and Methods section? Surely it should appear in the Results.
Figure 2 was moved to the “Results” section and the mention was removed from the methods section to prevent misunderstandings.
- Description of Fig. 2 in the main text and the figure legend should be more precise.
The description was adjusted in the main text and the figure legend was changed from:
“Figure 2 Box plot depiction of continuous hemoglobin levels (mmol/l) in study subjects according to the presence of symptoms (0 = no symptom; 1 = symptom)”
To:
“Figure 2: Box plot depiction of continuous hemoglobin levels (g/L) in study subjects according to the presence of the respective symptoms “fatigue”, “lack of energy”, “lack of concentration” and “dyspnea or weakness” (0 = symptom is not present; 1 = symptom is present)”
- Please change Hb concentration units to g/L throughout the manuscript as it is more common in the subject literature.
All Hb-concentrations were changed into g/L in the manuscript.
- I do not understand why the Authors decided to compare all participants (including participants with anemia) to participants with anemia in Table 3. In my view participants with and without anemia should be compared and p values for intergroup differences shown. Alternatively there should be 3 columns in the Table: all participants; participants without anemia; participants with anemia.
We added the column "participants without anaemia" and changed the percentages to the column percentages. Please note, that table 3 was not meant to be a comparison between the groups per se, but was included to describe the baseline characteristics of the study population. This is the reason that we do not include p-values in this table, which would additionally be of limited value because of the extent of some of the group size differences.
For the evaluation of differences in symptoms between the groups of participants with and without anemia we kindly ask you to refer to the multivariate logistic regression analyses in table 5 as these are more robust in determining the differences between these groups, especially because we account for confounding factors like age, sex or heart failure in the logistic regression analyses, that would influence intuitive comparisons based on table 3.
- Percents of participants in both groups are missing at the top of the Table 3.
We added the percentages to the top of table 3.
- Legend for Table 4 requires correction.
We have added the following legend to Table 4:
Anemic subjects from the population-based study cohorts SHIP-START-2 and SHIP-TREND were categorized according to WHO normal values for hemoglobin levels in the respective patient groups “women”, “pregnant women”, and “men”.
WHO: World Health Organization, SHIP: Study of Health in Pomerania
- In line 144 please give Me (IQR) and p value for age in both groups.
We are not sure what is meant here, as line 144 in our version does not allow for this. If still necessary after the additional analyses we have provided, we kindly ask for additional information.
- Table 5 - please explain what 2 models of multivariate logistic regression was used. This should also be clarified in the Materials and Methods.
Logistic regression models as derived from the table were used to analyze the association between the anemia symptom as an outcome variable and as relevant characteristics as exposure variables. Because the variables „number of medications“ and „medication with fatigue as a side effect“ are associated, these could not be used in the same model. For this reason 2 multivariate logistic regression models where applied as depicted in Table 5. The first of which (model 1) with „number of medications“ and the other model (model 2) with „medication with fatigue as a side effect” as one of the exposure variables. We used stepwise selection to select the most suitable exposure variables for use in the models.
We added the information to the methods section of the manuscript (see details in the next section).
- Why are there empty cells?
For the symptoms ”fatigue”, “lack of energy” and “lack of concentration” the comorbidities “heart failure”, “COPD” and “asthma” were not included. Stepwise selection was used to select the most suitable variables in the models. This information was added to the methods section of the manuscript:
“We used stepwise selection to select the most suitable variables in the models.”
The variables “Number of medications” and “Medication with fatigue as a side-effect" are correlated with each other and can therefore not be included in one and the same model.
We included the following sentence to the method section of the manuscript:
“The variables “number of medications” and “medication with fatigue as a side-effect" cannot be included in the same model because they are correlated.”
- In the 'Comparison with scientific literature' there is no mention of papers showing different prevalence of symptoms of anemia in anemia of various severity. It is important to stress this as participants in the study were mostly mildly anemic. Citing the paper from 1966 is concerning me. The cited references should be recent publications - optimally from within the last 5 years.
Scientific literature regarding anemia symptoms is surprisingly scarce. This is also the reason that this analysis was performed. We have systematically reviewed the literature but have not identified any recent papers that specifically describe anemia symptomatology in population-based or primary care settings. We have added additional references describing symptomatic anemia albeit in clinical settings.
- In conclusions the Authors should stress that anemia in these cohorts was mostly mild, occasionally moderate.
We have added the following sentence to limitations to stress the relatively mild severity of anemia in the study population:
“When interpreting the results, it should be considered that most study participants had mild anemia.”
- The English language style should be improved in some parts of the manuscript. I suggest that the manuscript is read by a native speaker.
The manuscript was proofread by a native speaker. A mention was added to the “Acknowledgements” section.
Reviewer 2 Report
This is basically an epidemiological study.
This reviewer has two major concerns.
1st: I disagree with the general use of anemia throughout the study. Authors just compare symptoms with Hb levels. They provide absolutely no clue about iron stores, transferrin saturation, hepcidin etc. thus, no pathophysiology at all. It is known that mechanisms of anemia influence symptoms, and also the slow or rapid constitution of the iron store depletion.
2nd: I would have been mostly interested by subgrouping ages over 70, as aged patients are the most exposed in general to iron deficiency and anemia for diverse reasons of which lack of appropriate iron intake or absorption, cancer (MDS), autoimmunity, etc. 70 is rather young, compared to 90 or 100.
Author Response
- 1st: I disagree with the general use of anemia throughout the study. Authors just compare symptoms with Hb levels. They provide absolutely no clue about iron stores, transferrin saturation, hepcidin etc. thus, no pathophysiology at all. It is known that mechanisms of anemia influence symptoms, and also the slow or rapid constitution of the iron store depletion.
We agree that iron storage and other factors regarding iron metabolism may play an important role regarding symptomatology. The incorporation of these factors into our analysis could provide additional information, especially in individual subjects.
Because of the population-based nature of our analysis we are able to provide information about a broad sample of the population, but it would contradict the nature of the analysis to review individual causes of anemia in-depth. Additionally, we are limited to the analysis of laboratory values that have been performed in the SHIP-study analyses, which did not include hepcidin or transferrin saturation, but included ferritin.
We have added a sensitivity analysis to evaluate the influence of iron deficiency as indicated by low ferritin levels as a factor associated with the symptomatology. The results of this analysis were added in Supplement table B. The results of this additional analysis did not change the study results, but we found that iron deficiency as indicated by low ferritin levels was independently associated with concentration problems in the multivariate logistic regression analysis. We added this information to the results section of the manuscript and redacted the limitations section.
We added the following information to the limitations section:
“In this study we were limited to the analysis of laboratory values that were available in the SHIP-study, which included ferritin as an indicator of iron storage, but did not include hepcidin or transferrin saturation as additional indicators of iron metabolism.”
In this analysis we follow a pragmatic approach, as in general practice patients frequently present with the symptoms investigated in our study, which often leads to laboratory testing. In the ambulatory setting, because of cost and lack of clinical importance often initially only a small range of parameters including blood count und ferritin levels are analyzed. Especially hepcidin is not available in standard laboratory testing and not feasible for the high numbers of patients with anemia symptoms in primary care settings.
Because ferritin has been deemed to influence symptoms of iron deficiency and/or anemia, we have added a sensitivity analysis to determine the association with anemia symptoms. We have included the analysis in Supplemental file B. Iron deficiency was not associated with fatigue, lack of energy and dyspnea and/or weakness in the multivariate analysis, but was associated with lack of concentration.
We added the following sentence to the method section:
Iron deficiency was defined as serum ferritin <8 µg/l for women and <26 µg/l for men.
We added the following words to the results section:
In multivariate analysis, no significant association between anemia and lack of concentration was found (Table 5), but iron deficiency was significantly associated with lack of concentration with an OR of 1,65 (95%CI 1.24-2.19) (Supplement Table B).
- 2nd: I would have been mostly interested by subgrouping ages over 70, as aged patients are the most exposed in general to iron deficiency and anemia for diverse reasons of which lack of appropriate iron intake or absorption, cancer (MDS), autoimmunity, etc. 70 is rather young, compared to 90 or 100.
We agree, that people in older age are more likely to be afflicted by anemia for the reasons mentioned above. The symptomatology may also change with age.
Because of the relatively small number of subjects in our study population with anemia in the oldest age groups, we have decided against subgroup analysis. As the SHIP cohort ages and more data is gathered, we will be able to perform a subgroup analysis of the older participant groups in the future, as we agree that anemia may affect the patients of 90 or 100 years old differently than patients between 70 and 80.
Round 2
Reviewer 2 Report
none